# Ultra-fast single-crystal polymerization of large-sized covalent organic frameworks

Lan Peng[1,2], Qianying Guo[1,2], Chaoyu Song [3], Samrat Ghosh [4], Huoshu Xu[5], Liqian Wang[1,2], Dongdong Hu[6], Lei Shi [3], Ling Zhao[6], Qiaowei Li[5], Tsuneaki Sakurai[4], Hugen Yan [3], Shu Seki[4], Yunqi Liu[2,7] & Dacheng Wei [1,2✉]

In principle, polymerization tends to produce amorphous or poorly crystalline materials. Efficiently producing high-quality single crystals by polymerization in solvent remains as an unsolved issue in chemistry, especially for covalent organic frameworks (COFs) with highly complex structures. To produce μm-sized single crystals, the growth time is prolonged to >15 days, far away from the requirements in practical applications. Here, we find supercritical $CO_2$ (sc-$CO_2$) accelerates single-crystal polymerization by 10,000,000 folds, and produces two-dimensional (2D) COF single crystals with size up to 0.2 mm within 2~5 min. Although it is the fastest single-crystal polymerization, the growth in sc-$CO_2$ leads to not only the largest crystal size of 2D COFs, but also higher quality with improved photoconductivity performance. This work overcomes traditional concept on low efficiency of single-crystal polymerization, and holds great promise for future applications owing to its efficiency, industrial compatibility, environmental friendliness and universality for different crystalline structures and linkage bonds.

[1] State Key Laboratory of Molecular Engineering of Polymers, Department of Macromolecular Science, Fudan University, Shanghai, China. [2] Institute of Molecular Materials and Devices, Fudan University, Shanghai, China. [3] Department of Physics, Fudan University, Shanghai, China. [4] Department of Molecular Engineering, Graduate School of Engineering, Kyoto University, Nishikyo-ku, Kyoto, Japan. [5] Department of Chemistry, Fudan University, Shanghai, China. [6] School of Chemical Engineering, East China University of Science and Technology, Shanghai, China. [7] Institute of Chemistry, Chinese Academy of Sciences, Beijing, China. ✉email: weidc@fudan.edu.cn

It is generally considered that the growth of a single crystal is much more time-consuming than amorphous or polycrystalline materials[1]. Compared with other crystals, crystalline materials constructed via covalent bonds face a greater challenge in efficient and precise production, since the formation and breakage of strong covalent bonds have lower reversibility than other linkages[2–6]. Single-crystal growth normally requires harsh conditions such as high temperatures or pressures[7]. Under a mild condition of polymerization in solvent, the challenge becomes more critical[2,4,5,7]. To obtain high-quality single crystals, it is necessary to decelerate the polymerization and prolong the growth time[8,9]. Considerable long-time is consumed for the polymerization along with reversible error-correction of the covalent lattices[4,6,7,10–13]. Therefore, although covalent bond ensures higher stability of crystalline materials in the application, it brings single-crystal polymerization a puzzling contradiction between growth time and product quality, causing dilemma in not only researches but also applications of these materials[2,7,10,11]. As a type of crystalline porous polymers, covalent organic frameworks (COFs) are obtained by atomically precise integration of organic building blocks via strong covalent bonds in a highly ordered periodic manner[10,11,14–16]. They have highly complex structures with different topologies, pore structures, linkages, functional groups, etc. and the interaction strength between intra- and inter-layers of two-dimensional (2D) COFs differ by several orders of magnitude[3,10,11,14]. Owing to special well-defined stable structures together with tailored functionalities, they have attracted numerous interests since 2005 and promised widespread applications ranging from gas separation to photoelectronics[10,11,15–20]. However, such a highly complex structure, together with the fundamental limitation from single-crystal polymerization, gives rise to a great difficulty in the fast growth of high-quality samples, especially large-sized single crystals.

By solvothermal, ionothermal, mechanochemical, or microwave polymerization, only polycrystalline powders comprising disorderly aggregated crystallites smaller than 200 nm are obtained, even when the whole reaction lasts for 1–4 days[15,17–22], hampering state-of-the-art applications of single-crystalline COFs. Although numerous researchers attempt to accelerate the polymerization, the sample quality is normally poor with small grain sizes[5,23–25]. To obtain 1–2 μm-sized single crystals, it is required to decelerate the polymerization by decreasing the addition rate of monomers[8,26], and the time is prolonged to 15 days or longer. In industrial-scale production, long reaction time means high cost, low efficiency, and poor feasibility. Therefore, existing knowledge and technologies are still far away from the requirement of practical applications in efficiently and precisely producing covalent single crystals by polymerization. More understanding on single-crystal polymerization is required. Herein, we demonstrate the first example of ultra-fast single-crystal polymerization in solvent, and find the pivotal role of solvent medium in the process. Compared with traditional liquids, sc-$CO_2$ extremely accelerates nucleation, polymerization as well as the reversible error-checking and proof-reading. We develop a methodology, named supercritically solvothermal polymerization, and realize ultra-fast growth of high-yield COF single crystals with different crystalline structures and linkage bonds. The crystal size is up to 0.2 mm, ~100 times larger than 2D COF single crystals reported previously[8]. The growth rate reaches 40 μm min$^{-1}$, up to 100,000 times faster than the best results of growing 2D COF single crystals[8,9] and around 6000 times faster than ultra-fast polymerization technologies of producing polycrystalline or nanocrystalline COFs[23–25].

## Results

### Single-crystal polymerization in sc-$CO_2$.
Three types of 2D COFs (sc-COFs, Fig. 1, Supplementary Fig. 1) with different structures and linkage bonds are prepared in sc-$CO_2$ (80 °C, 8 MPa) with 0.25% (vol.) n-butyl alcohol (n-BuOH), including imine-linked COFs (COF$_{TP-Py}$, COF$_{TB-BA}$) and boronate ester-linked COFs (COF$_5$). These COFs are also synthesized by solvothermal polymerization (os-COFs) in 50% (vol.) n-BuOH and 50% (vol.) 1,2-dichlorobenzene (o-DCB) as a comparison. After 5 min growth in sc-$CO_2$, Fourier transform infrared (FT-IR) spectra (Supplementary Fig. 2) confirm the polymerization. Optical microscope (OM) and scanning electron microscope (SEM) images (Figs. 1c, d, Supplementary Figs. 3–5) images reveal that all of the as-grown sc-COFs have high-yield rod-like morphologies with a length above μm-scale, distinct from that of disorderedly aggregated os-COFs (3 days). In particular, most sc-COF$_{TP-Py}$ (5 min) are 10–40 μm in length (Supplementary Fig. 6). We also grow sc-COF$_{TP-Py}$ for 2 min, and the length reaches up to 30 μm (Supplementary Fig. 7). Powder X-ray diffraction (PXRD) patterns (Fig. 1e) of the sc-COFs present sharp diffraction peaks with the highest in small-angle region, indicating that all sc-COFs (5 min or 90 min) are highly crystalline. The peak positions are in good agreement with the simulated patterns of AA eclipsed stacking models, which match well with previous literatures[15,27,28] and PXRD of corresponding os-COFs (3 days). No diffraction peaks appear in PXRD of os-COFs (30 min) or no samples (os-COF$_5$, 90 min) are obtained, indicating that the crystallization in organic solvent is much slower than that in sc-$CO_2$.

### Crystalline structure.
Transmission electron microscope (TEM) images and selected area electron diffraction (SAED) patterns (Fig. 2a–j, Supplementary Figs. 8, 9, 10) reveal high yield and highly crystalline features of the rod-like sc-COFs. The lattice fringes are clearly observed along the length direction with an inter-lattice separation of 2.0 nm (sc-COF$_{TB-BA}$), 2.4 nm (sc-COF$_5$), 1.8 or 2.3 nm (sc-COF$_{TP-Py}$), which correspond to COF$_{TB-BA}$ (100), COF$_5$ (100), COF$_{TP-Py}$ (020) or COF$_{TP-Py}$ (110) planes, respectively, indicating the 2D COF layers are stacked along the length direction via [001]. Cross-section TEM images (Fig. 2b) show clear fringes of sc-COF$_{TB-BA}$ with corresponding six-fold-symmetric fast Fourier transform (FFT) patterns, thus the micro-pores of 2D sc-COFs are oriented parallel to the rods. Although some rods are curved owing to the flexibility, all locations of the rod are well crystalline, and the microporous channels with open ends are continuous over the entire rod region without domain boundaries or interfaces (Fig. 2g–j, Supplementary Figs. 11, 12). The same set of SAED patterns are collected from different locations, indicating the single-crystalline nature of the 2D sc-COF rods (Supplementary Fig. 13). The sc-COF$_{TP-Py}$ grown for 2 min also has the same feature with high-quality single-crystalline structure (Supplementary Fig. 14). In contrast, all os-COFs are disorderedly polycrystalline with nm-scale domains (Fig. 2k, l, Supplementary Fig. 15). Cross-polarized OM images (Fig. 2m, n, Supplementary Fig. 16) confirm single crystal-linity of the sc-COFs, as uniform polarized light extinction is observed over the entire length of the rods (Supplementary Video 1), different from os-COFs (Supplementary Fig. 17, Supplementary Video 2). Some sc-COF$_{TP-Py}$ crystals are larger than 0.1 mm, up to 0.20 mm, which are the largest 2D COF single crystals reported till now (Fig. 1b).

### Growth rate and crystal quality.
To study the growth rate, 2D COFs are grown for different times. SEM images (Supplementary Figs. 7, 18) show rod-like crystals with sizes up to several tens (sc-COF$_{TP-Py}$) of micrometers are obtained within 2–5 min, with similar morphologies and lengths to sc-COFs grown for 2 h and 12 h (Supplementary Fig. 19), indicating that most single-crystal polymerization finishes within 5 min owing to the consuming monomer. The os-COFs have distinct morphologies (Supplementary Fig. 20),

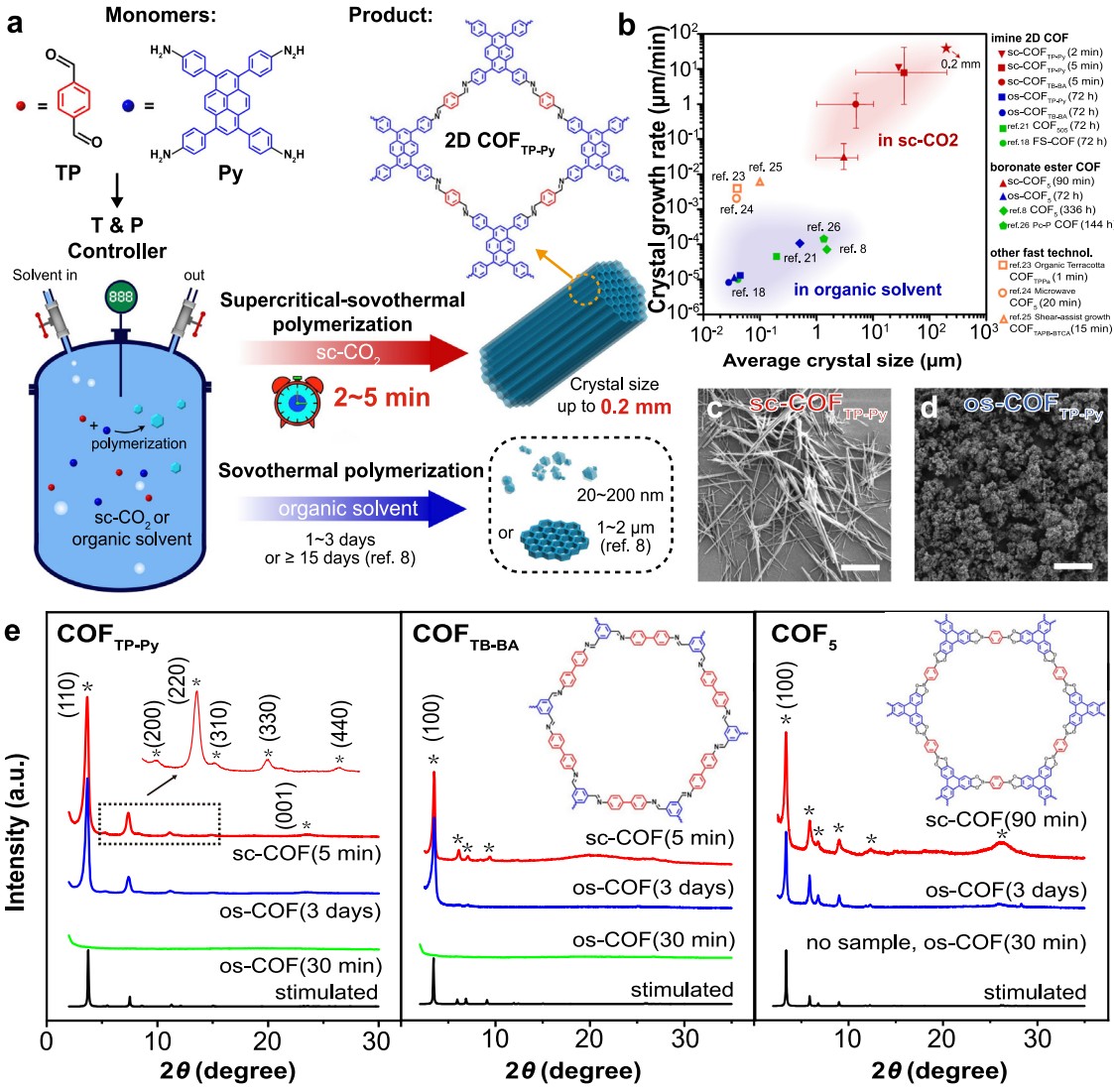

**Fig. 1 Supercritically solvothermal polymerization. a** Schematic of the COF$_{TP-Py}$ synthesis. **b** Crystal size and growth rate in this work (sc-COFs and os-COFs) compared with some best-reported results. The red star indicates the best result in sc-CO$_2$. **c, d** SEM images of sc-COF$_{TP-Py}$ (5 min) and os-COF$_{TP-Py}$ (3 days). **e** PXRD patterns of sc-COFs (5 or 90 min), os-COFs (3 days), os-COFs (30 or 90 min), and the simulated patterns. The characteristic peaks are marked by stars. The scale bars are 10 μm.

which are disorderedly aggregated with small crystallites, even if the growth lasts for 3 days. The crystal growth rate of sc-COFs reaches 40 μm min$^{-1}$ in sc-CO$_2$, up to 7 orders of magnitude faster compared with that of os-COFs (Fig. 1b).

The crystal quality is measured by PXRD. High-intensity diffraction peaks of sc-COF$_{TP-Py}$ appear after 2–5 min (Fig. 3a), which are similar to PXRD patterns of sc-COF$_{TP-Py}$ grown for 8–12 h and os-COF$_{TP-Py}$ grown for 3 days, indicating high-quality COFs are obtained in sc-CO$_2$ within minutes. Weak diffraction peaks of the solvothermal samples appear only after 2 h, indicating the crystallization in organic solvent is much slower. The full width at half maximum (FWHM) of the diffraction peak (Fig. 3b) is related to the crystal quality[29]. In sc-CO$_2$, FWHM of the sc-COF$_{TP-Py}$ (110) peak reaches 0.39° after 5 min, decreases slightly to 0.35° after 2 h, and maintains around 0.34° until 8 h. However, in organic solvent, FWHM of the os-COF$_{TP-Py}$ (110) peak is only 0.62° after 4 h. A time up to 1–3 days is required to obtain high-quality COFs with FWHM around 0.35°. Moreover, we use a mixture solvent of sc-CO$_2$ and n-BuOH (Fig. 3c). Higher sc-CO$_2$ content leads to a larger growth rate and smaller FWHM of COF$_{TP-Py}$ (110) peak, indicating that the solvent medium plays

a pivotal role in not only ultra-fast crystallization but also the formation of high-quality crystalline structures.

**Single-crystal polymerization mechanism.** Ultra-fast single-crystal polymerization is related to the special properties of sc-CO$_2$. In organic solvent, the COF crystals are obtained by initial polymerization and nucleation followed by a subsequent crystalline growth or transformation process[3,5,6,11–13]. Similarly, the growth of 2D COFs in sc-CO$_2$ involves extension of the lattice in the lateral [100] direction by covalent bond formation as well as in the vertical [001] direction by non-covalent stacking (Fig. 4)[12,13]. The assembly of monomers into the ordered framework by polymerization normally encounters many mistakes, such as overlays, defect sites, ill-defined agglomerations, interlayer cross-links, etc[4]. To obtain an ordered framework, a time-consuming reversible self-correction is required along with the polymerization for error-checking and proof-reading[4,6–13]. Considering the large surface tension (20–50 dyn cm$^{-1}$) and viscosity (0.3–4 cp, normally) of organic solvent, the transport of precursors

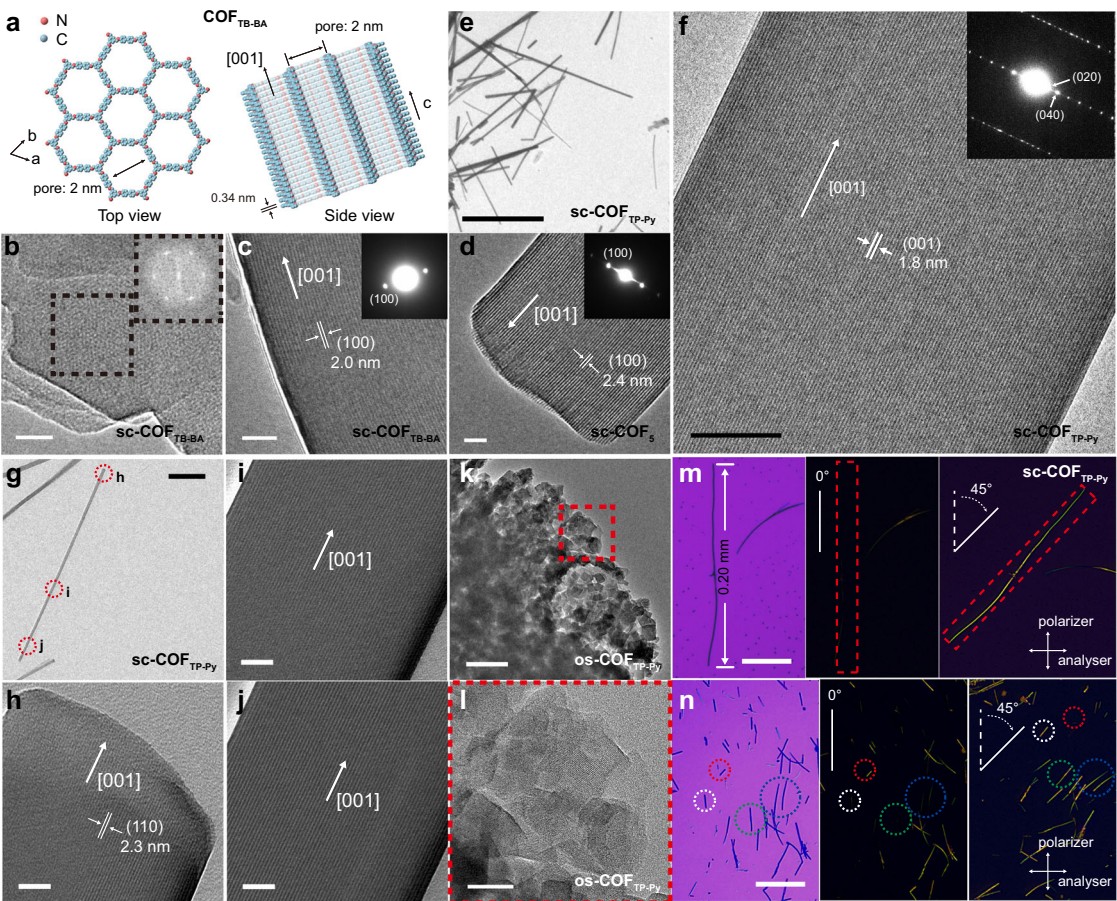

**Fig. 2 Characterization of the crystalline structure. a** Top and side view of COF$_{TB-BA}$. **b** Cross-section TEM image of sc-COF$_{TB-BA}$. The inset is the FFT pattern obtained from the dashed square. **c–g** TEM images of sc-COF$_{TB-BA}$ (**c**), sc-COF$_5$ (**d**), and sc-COF$_{TP-Py}$ (**e–g**). The insets show the corresponding SAED patterns. **h–j** High-resolution TEM images of a sc-COF$_{TP-Py}$ crystal taken from the regions marked by red circles in (**g**). **k, l** TEM and enlarged TEM images of os-COF$_{TP-Py}$. **m, n** OM and cross-polarized OM images of a 0.20 mm long sc-COF$_{TP-Py}$ crystal (**m**) and sc-COF$_{TP-Py}$ crystals (**n**). The scale bars are 20 nm in (**b–d**), (**h–j**), 50 nm in (**f, l**), 200 nm in (**k**), 2 μm in (**g**), 10 μm in (**e**), 50 μm in (**m**) and (**n**).

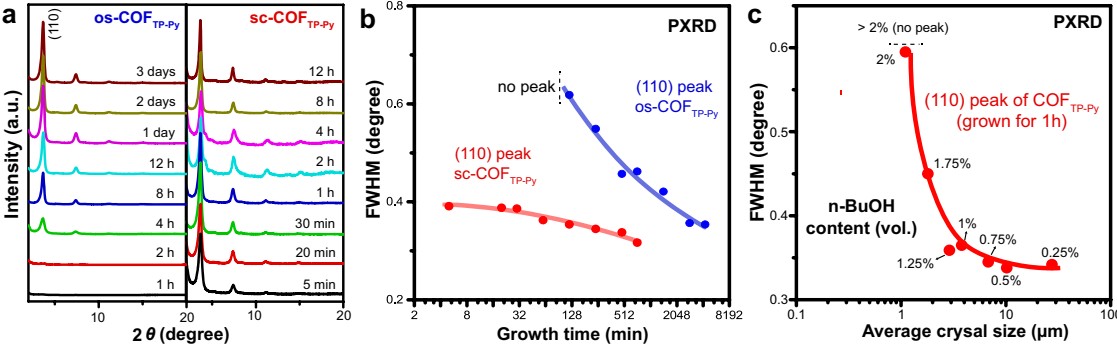

**Fig. 3 Ultra-fast crystal growth of sc-COFs. a** PXRD of the samples of os-COF$_{TP-Py}$ and sc-COF$_{TP-Py}$ grown for different times. **b** FWHM of sc-COF$_{TP-Py}$ (110) and os-COF$_{TP-Py}$ (110) peaks as a function of growth time. **c** FWHM of the COF$_{TP-Py}$ (110) peak and average crystal sizes of the products grown for 1 h in a mixture solvent of n-BuOH and sc-CO$_2$.

(including monomers and oligomers) and byproducts to or from the reaction sites is restricted by the slow diffusion rate in the solvent as well as in the micro-pores or interstices of COFs. As a result, the precursor concentration is low while the byproduct concentration is high near the reaction sites, which drastically slows down the reversible reaction, decelerates the framework formation, and brings a fundamental limitation in ultra-fast polymerization of high-quality single crystals.

Supercritical fluid technologies have been generally utilized in pharmaceutical, food, chemical manufacturing, and industries since 1960s[30], which realize a green, scalable and efficient chemical or biochemical synthesis and processing in a special medium distinct from normal gases and liquids[31–33]. Considering the inertness, sc-CO$_2$ functions as an inert solvent in the reactions[34], and the reaction thermodynamics is similar to that in liquid mediums. The difference is that the supercritical fluid

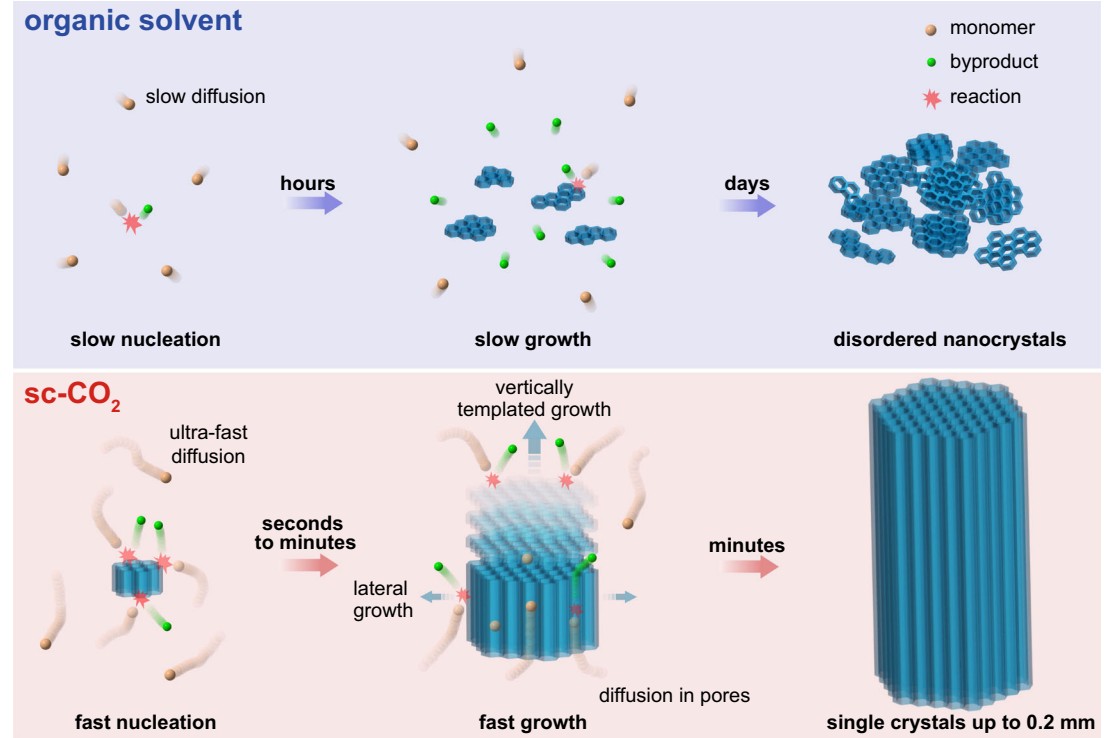

**Fig. 4 Single-crystal polymerization mechanism.** Schematic illustration of polymerization in organic solvent or sc-$CO_2$. In sc-$CO_2$, the ultra-fast diffusion of monomers, oligomers, and byproducts in medium and micro-pores accelerates polymerization as well as error-checking and proof-reading, while the suitable solubility accelerates vertically epitaxial growth, resulting in minutes-growth of single crystals.

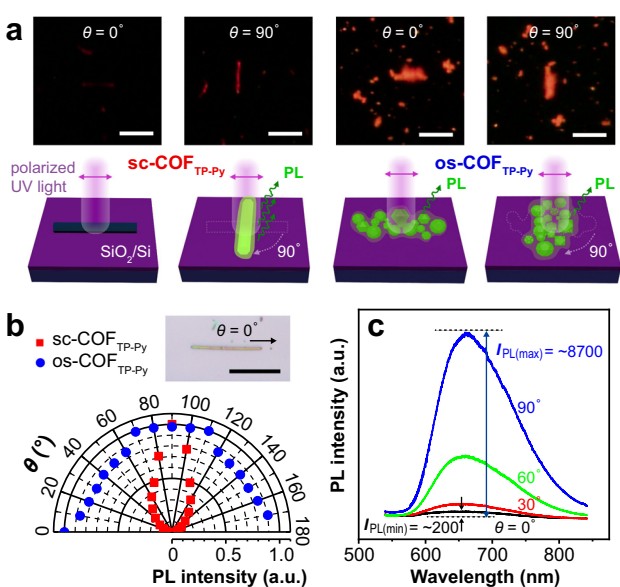

**Fig. 5 Polarized photoluminescence. a** Polarized fluorescence microscopy images and schematic illustration of sc-COF$_{TP-Py}$ and os-COF$_{TP-Py}$ at $\theta = 0°$ and 90°. **b** Angle-dependent PL intensity at different $\theta$. The inset is an optical microscopy image of a sc-COF$_{TP-Py}$ crystal. **c** PL spectra of the crystal when $\theta$ is 0°, 30°, 60°, 90°. The DR (defined as $I_{PL(max)}/I_{PL(min)}$) is calculated to be 43.5. The scale bars are 10 μm in (**a**), 20 μm in (**b**).

combines gas-like viscosity and surface tension with liquid-like density and solvating properties[35]. The viscosity of sc-$CO_2$ is about 0.02 cp (80 °C, 8 MPa), while the surface tension is near zero[36]. The diffusion coefficient is 1–2 orders of magnitude higher than that in organic solvent[37]. As a result, sc-$CO_2$ allows

increased penetrating ability and much faster diffusion of precursors and by-products in the medium and in the micro-pores or interstices of COFs[38]. It is expected that the precursors can be sufficiently supplied to the reaction sites while the byproducts be fast removed from there. Thus, sc-$CO_2$ greatly influences the reaction kinetics and accelerates the polymerization along with the error-checking and proof-reading process, resulting in ultra-fast nucleation and extension of lateral covalent frameworks. As an evidence, we grow 2D COFs in the nanometer-scale interstices of highly ordered 3D monolith made of 1 μm-sized polystyrene spheres (PSs). After removing PSs, a free-standing 3D-ordered structure of sc-COFs is obtained (Supplementary Figs. 21, 22). As a comparison, in 1,4-dioxane and mesitylene mixture, no COFs can grow in the interstices of the 3D monolith (Supplementary Fig. 23). This result indicates high diffusion and penetrating ability in sc-$CO_2$. To obtain an eclipsed non-covalent stacking structure of 2D COFs, a new layer forms via oligomer stacking and subsequent lateral template polymerization on top of an old layer[13]. The solubility of supercritical fluid is intermediate between that of a gas and a liquid[33]. The moderate solubility of sc-$CO_2$ allows nucleation and efficient stacking of oligomers vertically on old layers (Supplementary Fig. 24), resulting in the growth of the crystal along [001] direction.

To prove the mechanism, control experiments tune solvent property by adding n-BuOH in sc-$CO_2$. With higher n-BuOH content, the mixture solvent has lower diffusion ability and larger solubility. As a result, the products (1 h) have reduced crystal length (Supplementary Figs. 25–27) and increasing FWHM of COF$_{TP-Py}$ (110) peak (Fig. 3c), when n-BuOH content increases from 0.25% (vol.) to 2% (vol.). Finally, the products become amorphous without any diffraction peaks when the content is >2% (vol.). This result indicates that the solvent medium plays a pivotal role in not only ultra-fast growth but also the formation of

high-quality crystalline structures. Therefore, fast revisable reaction as well as efficient epitaxial nucleation and growth of the new layers in sc-$CO_2$ realize minutes-growth of COFs into large-sized rod-like single crystals.

**Polarized photoluminescence**. We measure the photoluminescence (PL) emission from sc-$COF_{TP-Py}$ and os-$COF_{TP-Py}$ (Fig. 5a). Angle-dependent emission only occurs in the case of sc-$COF_{TP-Py}$ (5 min). When the polarization angle ($\theta$) changes from 0° to 180° (the long axis direction is set to 0°), the PL emission ($I_{PL}$) from the crystal (Fig. 5b) gradually becomes stronger (0°–90°), and then becomes weaker (90°–180°). The dichroic ratio (DR) (defined as $I_{PL}$ maximum/$I_{PL}$ minimum) is as high as 43.5, and the polarization ratio ($\rho$)[39], defined as $(DR - 1)/(DR + 1)$, is 0.96 (Fig. 5c). In the case of os-$COF_{TP-Py}$ (3 days) or COFs produced by other methods, the $\rho$ is only ~0 (Fig. 5b). It is the first report of polarized PL emission from COFs. The $\rho$ is among the highest values of organic or polymeric crystalline materials[40], which verifies not only single-crystalline nature but also ultra-high crystalline quality of sc-$COF_{TP-Py}$ (see Supplementary Note 1)[41].

**FP-TRMC photoconductivity**. In our measurement, the spatial size of the oscillating motion of charge carriers is estimated within several nanometers at a maximum[42]. Thus, the grain and/or domain boundary will not affect the $\phi\Sigma\mu$ obtained by FP-TRMC. The FP-TRMC result is not related to the grain size but the actual quality (or crystallinity) of the as-grown COFs[43].

We measure the samples by flash-photolysis time-resolved microwave conductivity (FP-TRMC) under $N_2$ atmosphere. In the measurement, charge carriers are generated upon photo excitation and the local motion of the carriers can be probed via dielectric loss of the low-power microwave probes. The observable region of charge carriers can be extended to a very short time region, which allows us to obtain the intrinsic charge-carrier mobility within several nanometers after laser pulse irradiation under a rapidly oscillating electric field[44,45]. This technique is devoid of grain boundaries, impurities or crystal size, which reflects long-range (>10 nm) carrier transport along the ordered structure in crystalline domains[46]. Thus, the FP-TRMC photoconductivity ($\phi\Sigma\mu$), where $\phi$ and $\Sigma\mu$ represent the photocarrier generation yield and the sum of the photogenerated charge carrier mobilities, respectively[46], cannot be affected by the length of the crystals, but gives an estimate of the actual crystallinity in nanometer-scale level (see Supplementary Note 2). After 5 min growth in sc-$CO_2$, the photoconductivity transients of sc-$COF_{TP-Py}$ show a rapid rise within the time constant of the present set of apparatus in current, with a maximum $\phi\Sigma\mu$ value of $5.8 \times 10^{-6}$ $cm^2$ $V^{-1}$ $s^{-1}$ at an excitation photon density of $1.8 \times 10^{16}$ photons $cm^{-2}$. $\phi\Sigma\mu$ reaches ~$6.8 \times 10^{-6}$ $cm^2$ $V^{-1}$ $s^{-1}$ after growth in sc-$CO_2$ for 20 min to 12 h (Fig. 6). As a comparison, $\phi\Sigma\mu$ of os-$COF_{TP-Py}$ cannot reach the same value, even after growth for 3 days. Therefore, ultra-fast polymerization in sc-$CO_2$ not only obtains large-sized single crystals, but also results in comparable or higher crystal quality compared with os-COFs grown for days.

## Discussion

Practical application of COFs received limited success due to challenges in synthesizing high-quality products rapidly, industry-compatibly and environmentally friendly. Supercritically solvothermal polymerization reduces the time needed for growing large-sized 2D COF single crystals from months to minutes, which is hard or even impossible previously. Despite ultra-fast growth rate, it increases the crystal size up to 0.2 mm (sc-$COF_{TP-Py}$), 2 orders of magnitude

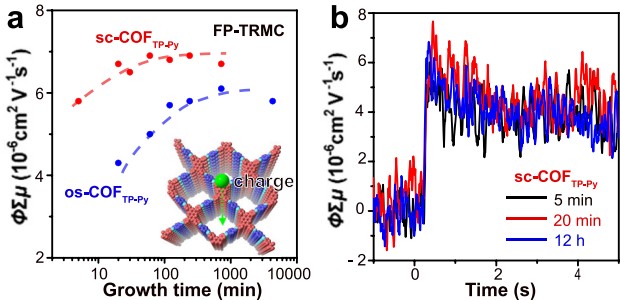

**Fig. 6 FP-TRMC measurement. a** Photoconductivity ($\phi\Sigma\mu$) of sc-$COF_{TP-Py}$ and os-$COF_{TP-Py}$ as a function of the growth time. **b** FP-TRMC transients of the sc-$COF_{TP-Py}$ grown for 5 min, 20 min and 12 h, respectively.

larger than the largest single crystals of 2D COFs reported previously[8], and also improves the quality and photoconductivity performance. Sc-COFs can be further purified and activated in sc-$CO_2$ (Supplementary Fig. 28). Although the solubility of sc-$CO_2$ isn't so strong as organic solvents, which results in a few of uncleaned monomers or oligomers in the nanochannels of 2D COFs, it is impressive that the surface area still reaches 1710 $m^2$ $g^{-1}$, comparable to that of os-$COF_{TP-Py}$ (1947 $m^2$ $g^{-1}$). Thus, the entire procedure is environmentally friendly without using amounts of organic solvents. Considering these as well as wide usage of sc-$CO_2$ in industry, it solves all the above challenges with universality for different linkages of 2D COFs, which greatly shortens the gap between synthesis and practical applications. Moreover, obstacle in synthesis hampers researches and applications of 2D COF single crystals. This work solves the problem and finds that the high quality and single-crystalline nature of sc-COFs give rise to highly polarized PL, showing that the 2D COF single crystals produced in sc-$CO_2$ have great potential in special application fields which previous poly-crystalline samples haven't. It is worth noting that there is a general belief that slow crystallization is necessary to grow high-quality single crystals[1]. In contrary, this result indicates ultra-fast polymerization in sc-$CO_2$ can produce single crystals with both larger size and higher quality, compared with slow crystallization, overcoming the contradiction between growth time and product quality. This work provides a new understanding of ultra-fast single-crystal polymerization in supercritical liquid. Besides boronate ester and imine 2D COFs, it also holds great promise in the efficient and precise construction of various COFs and other covalent crystalline materials, opening up other opportunities for not only fundamental research but also practical application of these materials and their single crystals.

## Methods

**Supercritically solvothermal polymerization**. (1) sc-$COF_{TP-Py}$: A mixture of terephthalaldehyde (TP, 5.4 mg, 0.04 mmol) and 1,3,6,8-tetra(4'-aminophenyl) pyrene (Py, 11.3 mg, 0.02 mmol) was dissolved in 100 μL n-butyl alcohol (0.25% vol., as a cosolvent) (n-BuOH) and 100 μL acetic acid, then transferred to a 40 mL stainless steel reactor. The system was charged with 8 MPa $CO_2$, heated to 80 °C, and reacted for 5 min. After the reaction, the reactor was slowly depressurized at a rate of 1–2 MPa $min^{-1}$. The precipitate was collected by filtration, washed with acetone, tetrahydrofuran, and dried in a vacuum oven at 100 °C.

(2) sc-$COF_{TB-BA}$: A mixture of 1,3,5-triformylbenzene (TB, 9.7 mg, 0.06 mmol) and 4,4'-biphenyldiamine (BA, 16.6 mg, 0.09 mmol) was dissolved in 100 μL n-BuOH and 100 μL acetic acid, and transferred to a 40 mL stainless steel reactor. The system was charged with 8 MPa $CO_2$, heated to 80 °C, and reacted for 5 min. After the reaction, the reactor was slowly depressurized at a rate of 1–2 MPa $min^{-1}$. The precipitate was collected by filtration, washed with acetone, tetrahydrofuran, and dried in vacuum oven at 100 °C.

(3) sc-$COF_5$: A mixture of 2,3,6,7,10,11-hexahydroxytriphenylene (HHTP) (11 mg, 0.034 mmol) and benzene diboronic acid (BDBA) (8.28 mg, 0.05 mmol) was dissolved in 200 μL n-BuOH, and transferred to a 40 mL stainless steel reactor and the system was heated to 90 °C and pressurized up to 9 MPa $CO_2$, reacted for 90 min. After the reaction, the reactor was slowly depressurized at a rate of 1–2 MPa $min^{-1}$. The precipitate was collected by filtration, washed with acetone, and dried in vacuum overnight.

**Solvothermal polymerization.** (1) os-COF$_{TP-Py}$: A mixture of terephthalaldehyde (TP, 5.4 mg, 0.04 mmol) and 1,3,6,8-tetra(4′-aminophenyl) pyrene (Py, 11.3 mg, 0.02 mmol) was dissolved in 0.5 mL n-BuOH and 0.5 mL1,2-dichlorobenzene (o-DCB), and transferred to a reaction tube. After sonication for 5 min at room temperature, the mixture was added 0.1 mL 6 M aqueous acetic acid. Then the tube was degassed through three freeze-pump-thaw cycles, sealed under vacuum and heated at 120 °C for 3 days. After the reaction, the precipitate was collected by filtration, washed with acetone, tetrahydrofuran, and dried in vacuum oven at 100 °C[47].

(2) os-COF$_{TB-BA}$: A mixture of 1,3,5-triformylbenzene (TB, 9.7 mg, 0.06 mmol) and 4,4′-biphenyldiamine (BA, 16.6 mg, 0.09 mmol) was dissolved in 0.5 mL mesitylene and 0.5 mL dioxane, and transferred to a reaction tube. After sonication for 5 min at room temperature, the mixture was added 0.1 mL 6 M aqueous acetic acid. Then, the tube was degassed through three freeze-pump-thaw cycles, sealed under vacuum and heated at 120 °C for 3 days. After reaction, the precipitate was collected by filtration, washed by acetone, tetrahydrofuran and dried in vacuum oven at 100 °C[27].

(3) os-COF$_5$: A mixture of 2,3,6,7,10,11-hexahydroxytriphenylene (HHTP) (11 mg, 0.034 mmol) and benzene diboronic acid (BDBA) (8.28 mg, 0.05 mmol) was dissolved in 0.5 mL mesitylene and 0.5 mL dioxane, and transferred to a reaction tube. After sonication for 5 min at room temperature. Then the tube was degassed through three freeze-pump-thaw cycles, sealed under vacuum, and heated at 100 °C for 3 days. After the reaction, the precipitate was collected by filtration, washed with acetone and dried in vacuum overnight[48].

**Characterization.** The samples were measured by PXRD (PXRD, Bruker, D8). All the COFs were recorded in the 2θ range between 2° and 40°. The radiation was Cu Kα ($\lambda$ = 1.54 Å), and the data collection was carried out using an Aluminum holder at a scan speed of 1° min$^{-1}$ and a step size of 0.02°. Fourier transform infrared (FT-IR) spectra were collected using a thermofisher Nicolet 6700 spectrometer.

TEM images were collected using FEI Tecnai G2 F20 S-Twin (acceleration voltage: 200 kV). SEM images were collected using Zeiss Gemini SEM500. The SEM samples were prepared by evaporating a drop of alcohol with COFs on a clean Si/SiO$_2$ wafer. OM and cross-polarized OM images were collected using a polarized OM (Leica DM2500P).

The N$_2$ adsorption-desorption experiments were conducted on an Autosorb iQ$_2$ (Anton Paar) surface area analyser. The os-COF$_{TP-Py}$ sample was washed in THF by Soxhlet's extraction for 3 days, dried in vacuum for 12 h, and degassed at 120 °C for 12 h before the measurement. The sc-COF$_{TP-Py}$ sample was purified by liquid CO$_2$ for 5 times, kept in sc-CO$_2$ (40 °C 8 MPa) for one hour (Samdri®-PVT-3D, Tousimis) and degassed at 0.13 MPa min$^{-1}$ before the measurements[49]. N$_2$ isotherms were recorded at 77 K by using ultra-high purity N2 (99.999% purity). The surface area was determined using Brunauer-Emmett-Teller (BET) adsorption model.

**Growth and characterization of 3D-ordered sc-COFs.** After being treated with aqueous hydrogen peroxide and concentrated sulfuric acid (vol./vol. = 3:1), Si/SiO$_2$ wafer was cleaned with isopropanol, acetone, and DI water, respectively. Then, the wafer was blown dry by a N$_2$ gun and suspended vertically in 10% wt. aqueous polystyrene (PS) microspheres dispersion (Thermo Corp.). The dispersion was placed in an oven at 40 °C overnight. By controlling the evaporation rate, highly ordered 3D monoliths of PS were obtained on the SiO$_2$/Si wafer[50].

To grow the sc-COF$_{TB-BA}$ in the highly ordered 3D monolith of PS, 16 mg 1,3,5-benzenetricarboxaldehyde and 27.6 mg 4,4′-diaminobiphenyl were mixed within 0.1 mL n-BuOH and 0.1 mL acetic acid, transferred into stainless steel reactor, then a piece of PS colloid crystal was immerged into the solution. Then the reactor was heated to 80 °C and charged with 8 MPa CO$_2$. After a reaction for 12 h, the reactor was cooled to room temperature and slowly depressurized at a rate of 1–2 MPa min$^{-1}$. The wafer was washed with THF and acetone, and characterized by SEM.

**Angle-dependent photoluminescence measurement.** The angle-dependent photoluminescence (PL) spectra were measured by a Horiba iHR550 spectrometer equipped with a silicon CCD detector array[51]. A 532 nm laser (spot size ∼1 μm, laser power of 70 μW) was used as the excitation source. The incident laser was focused by a Nikon Eclipse Ti microscope with a 50× (NA = 0.7) objective. In the measurements, the polarization direction of the incident laser light was parallel to the long axis of COF crystals. A linear polarizer was placed in the collecting optical path to detect the angle dependence of the PL emission. The polarized PL images were collected using Olympus BX-51-P fluorescence microscope at different polarization angles (by rotating the stage). We used a filter set which provided a narrow wavelength range (450 to 490 nm) for excitation, and allowed that emitted light at wavelengths above 515 nm passed to the detector. All optical experiments were performed in ambient environment and at room temperature.

**Flash photolysis time-resolved microwave conductivity test.** Flash photolysis time-resolved microwave conductivity (FP-TRMC) was performed by in situ TRMC system[52]. A high degree of sensitivity in the conductivity measurement was realized in a resonant cavity. The resonant frequency and power of microwave probes were set at ∼9.1 GHz and 3 mW, respectively. Charge carriers were generated upon direct excitation of solid samples with third-harmonic generation ($\lambda$ = 355 nm) from a Spectra-Physics model INDI-HG Nd: YAG laser (Continuum Inc., Surelite II, 5–8 ns pulse duration, 10 Hz), and the conductivity transients were recorded by a digital oscilloscope (Tektronix, TDS 3032B). All the experiments were performed at 25 °C in nitrogen.

## Data availability

All data supporting the findings of this work are available within this paper and its Supplementary Information (figures, videos). Other data are available from the corresponding author upon request.

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

## Acknowledgements

We thank Songhai Xie, Xiali Zhang, Qingsong Wu from Fudan University for their aid in the TEM test. This work was supported by the National Key R&D Program of China (2021YFE0201400, 2018YFA0703200), National Natural Science Foundation of China (51773041, 61890940, 21603038), Shanghai Committee of Science and Technology in China (18ZR1404900), the Strategic Priority Research Program of the Chinese Academy of Sciences (XDB30000000) and Fudan University.

## Author contributions

D.W. obtained ideas, designed research, and supervised the project. L.P., Q.G., D.H. and L.Z. prepared the samples. L.P. and L.S. produced 3D-ordered structures of COFs. L.P. and L.W. did SEM. L.P. and D.W. did TEM, POM, FT-IR, and XRD. H.X. and Q.L. measured BET. L.P., C.S. and H.Y. measured angle-dependent PL. S.G., T. S. and S.S. measured FP-TRMC. D.W., L.P. and Y.L. prepared the manuscript. All authors commented on the manuscript.

## Competing interests

The authors declare no competing interests.
