## [Peer Review File · Nature Communications]

REVIEWER COMMENTS

Reviewer #1 (Remarks to the Author):

This is an m/s on the topic of "Ultra-Fast Single-Crystal Polymerization of Large-Sized Covalent Organic Frameworks". I think this m/s has been written well, and it showcases a new mode of COF crystal growth, which is different from the other reports on COF crystal growth. Hence, I would recommend acceptance of this work per minor revision.

1. Since this work describes the COF crystal growth in supercritical CO₂ conditions, I was wondering what could be possible sources of energy for the C=N or other covalent bond formation? In another way, how essential polymerization is happening.
2. I would request authors to showcase the mechanism of the crystal growth concerning the 100 and 001 faces. Through authors have explained, I still could not understand how the 2D crystallization happens. How does supercritical CO₂ help the growth in the 2D plane?
3. If possible, some electron diffraction data could be provided. I think figure 2 could be modified a bit, and a more mechanistic model could be provided. Right now, the ED reflections data looked to be merged.
4. Also, do these sc-CO₂ based COFs showcase a better surface area and porosity? It would be an interesting fact to note.
5. I was also curious if this process will also work on Tp-based chemically stable COFs like TpPa-1, TpPa-2 etc.

Reviewer #2 (Remarks to the Author):

This manuscript reports a new methodology, named supercritically solvothermal polymerization, and realizes ultra-fast growth of high-yield COF single crystals with different crystalline structures and linkage bonds. The crystal size is up to 0.2 mm and the growth rate reaches 40 $\mu\text{m min}^{-1}$. The crystal size and growth rate have been tremendous improvement compared with 2D COF single crystals reported previously. In addition, the largest crystal size of 2D COFs exhibit higher quality with improved photoconductivity performance. This work breaks through the limitations of the traditional methods for growing big COF crystal and would impact COF research profoundly. I think it merits to be published in the journals after minor revision:

1. If possible, I am curious to know whether the SC-CO₂ method is viable for improving the crystallization of 3D-COF?
2. In Supplementary Fig. 27, why the BET surface area of high crystalline sc-COFTP-Py is smaller than that of polycrystalline os-COFTP-Py?
3. The length of the COF materials prepared by the method should be further discussed with more samples for more accurate statistics.
4. Figure 4 shows the single-crystal polymerization mechanism. Could the authors make it more clearly for easily understanding?

5. The photoconductivity of the single crystal COF based on different lengths should be further explored. What plays an important role for the improved photoconductivity performances, the preparation methods or the different length of the material?
6. Some mistakes are existed in the literature, such as "Nat Commun and J. mater. Chem. A.", etc.

For Reviewer #1:

“This is an m/s on the topic of "Ultra-Fast Single-Crystal Polymerization of Large-Sized Covalent Organic Frameworks". I think this m/s has been written well, and it showcases a new mode of COF crystal growth, which is different from the other reports on COF crystal growth. Hence, I would recommend acceptance of this work per minor revision.”

Reply: Thanks for your useful comments on how to revise the manuscript. According to your suggestion, we revised the manuscript as follows.

1) We simplified Figure 4 and modified the discussion of the mechanism, and made it clearer for the readers.

2) We added the enlarged SAED patterns of the sc-COFs (Supplementary Fig. 10).

3) We discussed the surface area of the sc-COFs grown in sc-CO₂.

4) We recounted the length of more COF crystals for more accurate statistics. The results still support the previous conclusion that most single-crystal growth finishes within 5 min (Supplementary Fig. 19).

5) Other revisions: we modified the grammar and the description, and provided 4 new literatures.

In the revised manuscript, we have addressed your comments carefully and made corresponding revisions marked in blue color which hope meet with your approval. The following are the point-to-point replies to your comments.

Q1: *“Since this work describes the COF crystal growth in supercritical CO₂ conditions, I was wondering what could be possible sources of energy for the C=N or other covalent bond formation? In another way, how essential polymerization is happening.”*

Reply: Thanks for your comment.

Supercritical fluids (SCFs) are a kind of solvent with unique physical and transport properties that are intermediate between those of a liquid or a gas. Thus, SCFs have been widely used as the media for chemical reaction and polymerization in last decades (*Chem. Rev.* **1999**, 99, 543). Considering the inertness, sc-CO₂ only functions as an inert solvent in these reactions (*Science* **2011**, 332, 835). In another word, the sc-CO₂ is not the energy source of the polymerization, and the reaction thermodynamics in sc-CO₂ is similar to that in organic solvents. The difference is that the sc-CO₂ can accelerate the mass transport of the reactants and byproducts, which influences the reaction kinetics and results in faster polymerization and crystallization of COFs compared with that in organic solvents.

According to your comment, we cited a new literature (ref. 34) and revised the manuscript to clarify this issue.

Q2: *“I would request authors to showcase the mechanism of the crystal growth concerning the 100 and 001 faces. Through authors have explained, I still could not understand how the 2D crystallization happens. How does supercritical CO₂ help the growth in the 2D plane?”*

Reply: Thanks a lot. In general, the growth of 2D COFs involves extension of the lattice in the lateral direction (100) by covalent reaction as well as in the vertical direction (001) by non-covalent stacking.

Different from traditional organic solvent, sc-CO₂ combines gas-like viscosity and surface tension with liquid-like density and solvating properties, which allows increased penetrating ability and much faster diffusion of precursors and by-products in the medium and in the micro-pores or interstices of COFs. The diffusion coefficient is 1~2 orders of magnitude higher than that in organic solvent (*J. Electrochem. Soc.* **2020**, 167, 054510). The precursors can be sufficiently supplied to the reaction sites while the byproducts be fast removed from there. Thus, sc-CO₂ overcomes the limitation of traditional organic solvents in mass transport, greatly accelerates nucleation, polymerization and reversible error-correction, and results in fast extension of the crystalline frameworks in the 2D plane.

To obtain an eclipsed non-covalent stacking structure, a new layer form via oligomer stacking and subsequent lateral template polymerization on top of an old layer (*Phys. Chem. Chem. Phys.* **2017**, 19, 9745). The solubility of supercritical fluid is intermediate between that of a gas and a liquid. The moderate solubility of sc-CO₂ allows nucleation and efficient stacking of oligomers on the old layers, resulting in growth of the crystal along the [001] direction.

In the revised manuscript, we revised the discussion of the mechanism and modified Figure 4. Moreover, our experimental data fully support the mechanism and exhibit high reliability and universality of ultra-fast single-crystalline polymerization of 2D COFs in sc-CO₂.

Q3: “If possible, some electron diffraction data could be provided. I think figure 2 could be modified a bit, and a more mechanistic model could be provided. Right now, the ED reflections data looked to be merged.”

Reply: Thanks for your suggestion.

In fact, the old version of the manuscript has provided electron diffraction data of sc-COF_{TB-BA}, sc-COF₅ and sc-COF_{TP-Py} in Fig. 2c, 2d, 2f and Supplementary Fig. 8, which show the high crystallinity of the as-grown samples. Moreover, we provided the SAED patterns of sc-COF_{TP-Py} collected from different locations of one single crystal. The same set of patterns are obtained, proving the single crystalline feature in another way.

Regarding your suggestion, we revised the mechanistic model so that it could be easier to be understood. Also, we provided original SAED data in the supplementary Fig. 10 and marked the lattice planes which should be clearer for the readers.

Q4: “Also, do these sc-CO₂ based COFs showcase a better surface area and porosity? It would be an interesting fact to note.”

Reply: The main point of this work is the fast preparation of 2D COF single crystals in sc-CO₂. To demonstrate the advantages, we prepared and purified COF_{TP-Py} in sc-CO₂, avoiding large amount usage of organic solvent. After the organic solvent-free process, we study the surface area of the as-prepared sc-COF_{TP-Py}.

According to Supplementary Fig. 28, the BET surface area is $1710 \text{ m}^2 \text{ g}^{-1}$.

Although the solubility of sc-CO₂ isn't so strong as organic solvents, resulting in a few of uncleaned monomers or oligomers in the nanochannels of sc-COF_{TP-Py}, the surface area is still comparable to that of os-COF_{TP-Py} ($1947 \text{ m}^2 \text{ g}^{-1}$). As a comparison, other fast-preparation of 2D COFs improves the efficiency at the cost of the porosity. For instance, the surface area of mechano-chemically prepared Tp-based 2D COFs is $537 \text{ m}^2 \text{ g}^{-1}$ (*J. Am. Chem. Soc.* **2013**, 135, 17853), and that of the light-promoted grown hcc-COFs is only $598 \text{ m}^2 \text{ g}^{-1}$ (*Commun. Chem.* **2019**, 2, 60) after purification in organic solvents. Thus, it is impressive that high surface area and porosity of 2D COFs maintain after ultra-fast growth and organic solvent-free purification in sc-CO₂.

Besides, supercritical fluid technologies have been widely used in chemical industry. This work provides a scalable, environmental benign alternative synthetic route of high-quality 2D COF crystals. Considering the common usage of sc-CO₂ in post-synthesis activation (*Acc. Chem. Res.* **2010**, 43, 1166), this method can realize one-step synthesis and activation of 2D COFs, showing high potential in future applications.

In the revised manuscript, we made corresponding revisions.

Q5: *"I was also curious if this process will also work on Tp-based chemically stable COFs like TpPa-1, TpPa-2 etc."*

Reply: In fact, we are making efforts to produce other kinds of 2D COFs by supercritically solvothermal method, and we believe that it is possible to produce Tp-based COFs in sc-CO₂. Our recent results show that the synthesis parameters need be optimized considering features of different COFs. More detailed studies are required in the future to find out the most suitable conditions for Tp-based COFs. Right now, four different 2D COFs including boronate ester COFs and imine COFs have been prepared in this manuscript, which are adequate to prove the universality of this method in 2D COF synthesis. Thanks for your comment, and we are expecting to further develop the supercritically solvothermal method and report more COFs prepared in sc-CO₂ in the future.

For Reviewer #2:

“This manuscript reports a new methodology, named supercritically solvothermal polymerization, and realizes ultra-fast growth of high-yield COF single crystals with different crystalline structures and linkage bonds. The crystal size is up to 0.2 mm and the growth rate reaches 40 $\mu\text{m min}^{-1}$. The crystal size and growth rate have been tremendous improvement compared with 2D COF single crystals reported previously. In addition, the largest crystal size of 2D COFs exhibit higher quality with improved photoconductivity performance. This work breaks through the limitations of the traditional methods for growing big COF crystal and would impact COF research profoundly. I think it merits to be published in the journals after minor revision.”

Reply: Thanks for your comments. Your comments are helpful for us to improve the quality of this manuscript. According to your suggestion, we revised the manuscript as follows.

1) We simplified Figure 4 and modified the discussion of the mechanism, and made it clearer for the readers.

2) We added the enlarged SAED patterns of sc-COFs (Supplementary Fig. 10).

3) We discussed the surface area of the sc-COFs grown in sc-CO₂.

4) We recounted the length of more COF crystals for more accurate statistics. The results still support the previous conclusion that most single-crystal growth finishes within 5 min (Supplementary Fig. 19).

5) Other revisions: we modified the grammar and the description, and provided 4 new literatures.

In the revised manuscript, we have addressed your comments carefully and made corresponding revisions marked in blue color which hope meet with your approval. The following are the point-to-point replies to your comments.

Q1: *“If possible, I am curious to know whether the SC-CO₂ method is viable for improving the crystallization of 3D-COF?”*

Reply: Thanks for your suggestion. In fact, we are also wondering whether the sc-CO₂ method could be applied in ultra-fast production of 3D COFs. In this manuscript, we have explained about how sc-CO₂ helps the fast crystallization of 2D COFs. The special properties of sc-CO₂, combining both liquid-like solubility and gas-like viscosity, helps precursors and by-products diffusing in and out of micro-pores or interstices of 2D COFs with diffusion efficiency much faster than that in organic solvents. Thus, sc-CO₂ accelerates the crystallization of 2D COFs. In this perspective, this method is possible to improve the crystallization of 3D COFs.

As we know, there are some differences between 2D COFs and 3D COFs. Firstly, 2D COFs consist of discrete planar sheets with non-covalent interlayer stacking, whereas 3D COFs are interconnected and fully covalent systems. Secondly, the interlaced and interpenetrated 3D structure which proven to be ubiquitous in 3D COFs brings much less empty space for building blocks to reach the right place of the crystal lattice. Thirdly, the kinetic trapping by 3D networks often occurs during crystallization, owing to the bulkiness and rigidity of the monomeric precursors

impeding proximity between reactive groups, precluding further network rearrangement into the desired crystalline framework (*Commun. Chem.* **2018**, 1, 98). These differences bring greater difficulties for 3D COFs to crystallize in organic solvent as well as in sc-CO₂. Therefore, the growth of 3D COFs in sc-CO₂ is a different topic.

In this manuscript, we mainly focus on 2D COFs. 2D COFs including boronate ester and imine COFs have been prepared, which proves the universality of this method in 2D COF synthesis. We are expecting to further develop the supercritically solvothermal method and prepare 3D COFs in sc-CO₂ in the future.

Q2: “In Supplementary Fig. 27, why the BET surface area of high crystalline sc-COF_{TP-Py} is smaller than that of polycrystalline os-COF_{TP-Py}.”

Reply: In Supplementary Fig. 28 (revised version), the sc-COF_{TP-Py} is prepared and purified in sc-CO₂ without large amount usage of organic solvents. The solubility of sc-CO₂ is lower than that of organic solvents, thus the oligomer and unreacted monomers could not be thoroughly cleaned by sc-CO₂ from the nanochannels of sc-COF_{TP-Py}, leading to a little decline of the surface area. Nevertheless, the surface area reaches 1710 m² g⁻¹, comparable to 1947 m² g⁻¹ of os-COF_{TP-Py}.

As a comparison, other fast-preparation of 2D COFs improves the efficiency at the cost of the porosity. For instance, the surface area of mechano-chemically prepared Tp-based 2D COFs is 537 m² g⁻¹ (*J. Am. Chem. Soc.* **2013**, 135, 17853), and that of the light-promoted grown hcc-COFs is only 598 m² g⁻¹ (*Commun. Chem.* **2019**, 2, 60) after purification in organic solvents. Thus, it is impressive that the high surface area and porosity maintain after ultra-fast growth and organic solvent-free purification in sc-CO₂. Besides, supercritical fluid has been widely used in chemical industry. This work provides one-step, scalable, environmental benign synthesis and activation of 2D COF crystals, showing high potential in future applications.

In the revised manuscript, we added some discussion of the surface area.

Q3: “The length of the COF materials prepared by the method should be further discussed with more samples for more accurate statistics.”

Reply: According to your suggestion, we recounted the length of COF crystals with more samples. The results (Supplementary Fig. 19) still support the conclusion that most single-crystal polymerization finishes within 5 min.

Q4: “Figure 4 shows the single-crystal polymerization mechanism. Could the authors make it more clearly for easily understanding?”

Reply: Thanks for your kind suggestion. In the revised manuscript, we simplified Figure 4, so that it should be easier to be understood.

Q5: “The photoconductivity of the single crystal COF based on different lengths should be further explored. What plays an important role for the improved photoconductivity performances, the preparation methods or the different length of the material?”

Reply: Thanks for your suggestion.

Flash-photolysis time-resolved microwave conductivity (FP-TRMC) method measures the intrinsic photoconductivity within several nanometers of a material after laser pulse irradiation under a rapidly oscillating electric field. This technology has been widely utilized to quantitatively evaluate intrinsic charge-carrier transport of conjugated polymers, photo-sensitized materials, discotic liquid crystals, metal-containing organic compounds and so on (*Angew. Chem. Int. Ed.* **2012**, 124, 2672). It avoids the influence of factors such as interfacial defects, grain boundaries, etc., and the result is not related to the length of the crystals but the actual crystalline quality in several nanometers of the samples. Thus, in our work, the improved FP-TRMC photoconductivity comes from highly ordered crystal lattice instead of large size of the crystals.

In the revised manuscript, we added new literatures (*J. Phys. Chem. C* **2008**, 112, 16643; *Chem. Mater.* **2011**, 23, 4094) which clarify the mechanism of the FP-TRMC measurement.

Q6: "Some mistakes are existed in the literature, such as "Nat Commun and J. mater. Chem. A.", etc."

Reply: Thanks for the remind. We checked the manuscript and revised these mistakes accordingly.

REVIEWERS' COMMENTS

Reviewer #1 (Remarks to the Author):

I am satisfied with the changes and this m/s may be accepted as it is.

Reviewer #2 (Remarks to the Author):

I think the authors have made earnest efforts to address the comments on the previous version. The manuscript has been largely improved. I suggest accepting it for the journals.

Reviewer #1 (Remarks to the Author): “I am satisfied with the changes and this m/s may be accepted as it is.”

Reply: Thanks for your comment. We are very grateful for your aid which help us to improve this manuscript.

Reviewer #2 (Remarks to the Author): “I think the authors have made earnest efforts to address the comments on the previous version. The manuscript has been largely improved. I suggest accepting it for the journals.”

Reply: Thanks for your comment. We are very grateful for your aid which help us to improve this manuscript.